# Cetylpyridinium chloride and chlorhexidine show antiviral activity against Influenza A virus and Respiratory Syncytial virus in vitro

**Marina Rius-Salvador**[1☯]**, Maria Jesús García-Múrria**[1☯]**, Luciana Rusu**[2]**,
Manuel Bañó-Polo**[3]**, Rubén León**[3]**, Ron Geller**[2]**, Ismael Mingarro**[1]**, Luis Martinez-Gil**[1]*

**1** Departament de Bioquímica i Biologia Molecular, Institut Universitari de Biotecnologia i Biomedicina (BIOTECMED), Universitat de València, Burjassot, Spain, **2** Institute for Integrative Systems Biology (I2SysBio), UV-CSIC, Paterna, Spain, **3** Department of Microbiology, DENTAID Research Center, Cerdanyola del Vallès, Spain

☯ These authors contributed equally to this work.
* luis.martinez-gil@uv.es

**Data Availability Statement:** All relevant data are within the paper and its Supporting Information files.

## Abstract

### Background

The oral cavity is the site of entry and replication for many respiratory viruses. Furthermore, it is the source of droplets and aerosols that facilitate viral transmission. It is thought that appropriate oral hygiene that alters viral infectivity might reduce the spread of respiratory viruses and contribute to infection control.

### Materials and methods

Here, we analyzed the antiviral activity of cetylpyridinium chloride (CPC), chlorhexidine (CHX), and three commercial CPC and CHX-containing mouthwash preparations against the Influenza A virus and the Respiratory syncytial virus. To do so the aforementioned compounds and preparations were incubated with the Influenza A virus or with the Respiratory syncytial virus. Next, we analyzed the viability of the treated viral particles.

### Results

Our results indicate that CPC and CHX decrease the infectivity of both the Influenza A virus and the Respiratory Syncytial virus *in vitro* between 90 and 99.9% depending on the concentration. Likewise, CPC and CHX-containing mouthwash preparations were up to 99.99% effective in decreasing the viral viability of both the Influenza A virus and the Respiratory syncytial virus *in vitro*.

### Conclusion

The use of a mouthwash containing CPC or CHX alone or in combination might represent a cost-effective measure to limit infection and spread of enveloped respiratory viruses infecting the oral cavity, aiding in reducing viral transmission. Our findings may stimulate future

**Funding:** DENTAID partially funded this study and supplied mouthwash formulations. This work was supported by the Generalitat Valenciana (PROMETEO/2022/062) and grant PID2020-119111GB-I00 by MCIN/AEI/10.13039/501100011033. R.G. acknowledges funding from the European Commission – NextGenerationEU (Regulation EU 2020/2094), through the CSIC's Global Health Platform (PTI Salud Global). M.R-S is the recipient of a predoctoral contract from the Spanish Ministry of Science and Innovation (PRE2021-101042 by MCIN/AEI/10.13039/501100011033).

**Competing interests:** M.BP. and R.L. are researchers working for the DENTAID Research Center. The authors declare that no other competing interests exist.

clinical studies to evaluate the effects of CPC and CHX in reducing viral respiratory transmissions.

## Introduction

Respiratory viruses represent a severe risk to human health [1]. Seasonal influenza alone accounts for approximately 1 billion infections worldwide every year, including 3–5 million cases of severe illness and up to 650,000 deaths [2]. Respiratory syncytial virus (RSV) may cause severe bronchiolitis in infants. In addition to the pediatric burden of the disease, RSV is recognized as an important pathogen in older adults. The hospitalization and mortality rates among those aged 65 years and over approach the rates seen with influenza [3]. Furthermore, the repercussion of respiratory viruses on global health has been increasing in recent years. Not only new respiratory viruses are emerging but also the impact of old ones is augmented due to climate change as new conditions affect the biology of viruses, host susceptibility, and human behavior [4].

Numerous respiratory viruses, including influenza A and B viruses (IAV and IBV, respectively), parainfluenza virus, metapneumovirus, rhinovirus, adenovirus, and respiratory syncytial virus (RSV), replicate in the oral cavity [5–7]. The oral cavity constitutes the portal of entry for many of these viruses and serves as a site of replication, more importantly, is the source of droplets and aerosols that facilitate viral transmission. It is well-accepted that physical barriers applied to the oral cavity (e.g. facial mask) can reduce the spread of viruses and contribute to infection control. Additionally, the use of mouth rinses with antiviral properties has been shown to reduce symptoms and transmission of respiratory infections [5,8,9].

Cetylpyridinium chloride (CPC) is a mono-cationic quaternary ammonium compound consisting of a quaternary nitrogen ring connected to one or more hydrophobic side chains. CPC and other quaternary ammonium compounds have been used for decades against a variety of pathogens [10,11]. CPC has shown activity against Gram-positive and negative bacteria. Additionally, CPC has demonstrated antiviral activity against some enveloped viruses, including common human coronaviruses [12], Severe Acute Respiratory Syndrome coronavirus 2 (SARS-CoV 2) [13–16], Middle East Respiratory Syndrome virus (MERS) [17,18], and influenza virus [19]. Furthermore, mouthwash formulations containing CPC reduced cold symptoms in randomized placebo-controlled double-blind trials [20,21]. The positively charged "head" on CPC interacts with negatively charged lipids on the surface of biological membranes, both of viral or bacterial origin. On the other hand, the hexadecane tail integrates into the lipid membrane. At low concentrations, CPC affects the membrane osmoregulation and homeostasis. At high concentrations, CPC leads to the disintegration of the membranes [22].

Cationic polybiguanides, particularly chlorhexidine (CHX) in either of its forms (chlorhexidine gluconate or chlorhexidine acetate) have been used extensively as a disinfectant and antiseptic and are commonly included in cosmetic and pharmaceutical products such as eye drops, mouthwash, and toothpaste. Like CPC, CHX broad-spectrum antimicrobial effects are due to its ability to disrupt biological membranes. Precisely, the positively charged CHX molecule reacts with negatively charged phosphate groups on the surface of biological membranes destroying the integrity of the pathogen. Although CHX is primarily known for its bacteriostatic and bactericidal effects, it has also shown some antiviral effects [23]. Precisely, it has shown activity against Cytomegalovirus (CMV), Herpes simplex virus-1 (HSV-1), IAV, Hepatitis B

virus (HBV), Human Immunodeficiency virus-1 (HIV-1), and some coronaviruses including SARS-CoV-2, although its action against the latest remains controversial [9,15,24–26].

To provide additional measures to limit respiratory viral infection and transmission, we have tested *in vitro* whether CPC and/or CHX can decrease the infectivity of two of the most medically relevant respiratory viruses, IAV and RSV. We also tested the ability of commercial mouthwash preparations containing CPC or CHX to inhibit IAV and RSV infection *in vitro*. Our results indicate that the use of a mouthwash containing CPC or CHX alone or in combination might represent a cost-effective measure to limit infection and spread of enveloped respiratory viruses infecting the oral cavity, aiding in reducing viral transmission. Moreover, additional *in vivo* assays should be conducted to confirm our results.

## Materials and methods

### Influenza A virus infections

Madin-Darby canine kidney (MDCK) cells (American Type Culture Collection, Manassas, USA) were seeded on 24-well plates ($1.5x10^6$ cells/plate), maintained in Dulbecco's Modified Eagle Medium (DMEM) (Gibco-ThermoFisher Scientific, Waltham, USA) supplemented with 10% fetal bovine serum (FBS) (Gibco-ThermoFisher Scientific, Waltham, USA) at 37˚C with 5% CO2. After 24 hours, culture media was removed and cells were infected with the IAV strain IAV/WSN/33 at a multiplicity of infection (MOI) of 0.01 by adding 100 μL of the viral inoculum for 45 minutes. Next, the virus was removed and fresh culture media was added. After 48 hours, the viral load on the supernatants was measured by 50% Tissue Culture Infective Dose ($TCID_{50}$) on MDCK cells (96 well plates).

To test the effect of CPC, CHX IAV/WSN/33 was incubated for 2 minutes at room temperature with CPC (0.1%, 0.05%, 0.025%, and 0.0125% final concentration) or CHX (0.25%, 0.125%, 0.06%, and 0.03% final concentration) before the infection. A 20% stock solution of CPC in Phosphate-buffered saline (PBS) and a 20% CHX stock solution in $H_2O$ were diluted in PBS to a 2x working solution (that is 0.2%, 0.1%, 0.5%, and 0.025% of CPC or 0.5%, 0.25%, 0.125%, and 0.06% of CHX). Next, the working solutions were mixed 1:1 with a virus stock of $1x10^6$ plaque-forming units/mL (pfu/mL) of IAV/WSN/33 (DMEM, 10% FBS)). After treatment, the solutions were further diluted in PBS to eliminate the excess of the compounds and to obtain the desired virus concentration. We used sodium dodecyl sulfate (SDS) (Sigma, Burlington, USA) at 0.05% as a positive control and PBS as a negative control. CPC and CHX were provided by DENTAID S.L. Research Center (Cerdanyola, Spain).

To assess the effect of Vitis CPC Protect, Perio Aid Active Control (PAAC), and Perio Aid Intensive Care (PAIC) the $1x10^6$ pfu/mL of IAV/WSN/33 stock was mixed with the mouthwash solutions provided by DENTAID S.L.. at a 1:1 ratio. Therefore, the effective concentration of CPC and CHX is 0.035% CPC in Vitis CPC Protect, 0.025% CPC, 0.025% CHX in Perio Aid Active control (PAAC), and 0.025% CPC, 0.06% CHX in Perio Aid Intensive Care (PAIC) and the viral concentration $5x10^5$ pfu/mL. The virus and the mouthwash were incubated for 2 minutes. Next, the solutions were further diluted in PBS to eliminate the excess of the compounds and obtain the desired virus concentration.

### Respiratory syncytial virus infections

For the Respiratory syncytial virus (RSV) assays we use a variant that expresses the fluorescent protein mKate2 (RSV-mKate2) [27]. To generate this virus the modified RSV infectious clone system (BEI Resources, Manassas, USA) was transfected into HEK-293 cells together with a codon-optimized T7 polymerase (Addgene, Watertown, USA, 65974), and an RSV N, P, M2-1, and L expression plasmids at a ratio of 4:2:2:2:1 respectively. Transfections were done with

Lipofectamine 2000 (Thermo Fisher Scientific, Waltham, USA) following the manufacturer's protocol. Cells were transfected in 6 well plates and subsequently transferred to T25 flasks with Hep2 cells until cytopathic effect (CPE) was observed.

To assess antiviral activity of the analyzed compounds, $4 \times 10^4$ fluorescent forming units of RSV-mKate2 were incubated for 2 minutes at room temperature with the indicated concentration of CHX or CPC (that is, 0.2%, 0.1%, 0.5%, and 0.025% of CPC or 0.5%, 0.25%, 0.125%, and 0.06% of CHX) in a total volume of 50 µL of DMEM containing 2% FBS. The virus plus compound solutions were then diluted 250 fold in DMEM with 2% FBS to eliminate the excess of the CPC pr CHX and to obtain the desired virus concentration. Next, 0.1 mL of the treated virus was used to infect confluent Hep-2 cells (CCL-23, American Type Culture Collection, Manassas, USA) in 96-well plates for 24 hours. Finally, viral infection was assessed by quantifying the number of mKate2-positive cells using the Incucyte® SX5 Live-Cell Analysis System (Sartorious, Göttingen, Germany). The cell viability was analyzed by monitoring cell confluency using the Incucyte® SX5 Live-Cell Analysis System. CPC and CHX were provided by DENTAID. S.L.. Tests using the mouthwash solutions were performed similarly but using $4 \times 10^3$ fluorescent forming units of RSV-mKate2 and diluted 100-fold before infection of cells. PAAC, PAIC, and vehicle mouthwash solution were provided by DENTAID S.L..

## Analysis of cell viability

The toxicity of the CPC, CHX, and mouthwashes was assessed with CellTiter-Glo (Promega, Madison, USA) following the manufacturer's instructions. Briefly, MDCK or HeLa cells were seeded on white 96-well plates ($1 \times 10^4$ cells/plate) and treated with the aforementioned compounds at different concentrations. After 24 hours, 100 µL of CellTiter-Glo reagent was added to each well and the luminescence was measured using a Victor X3 Multimode Plate Reader (Perkin Elmer, Shelton, USA).

## Statistical analysis

The level of significance (p-value) when comparing the PBS treatment (gray bar) vs. any of the treatments was calculated using a two-tailed homoscedastic t-test. The null hypothesis ($H_0$) assumes $\mu$ = average viral titer of PBS-treated samples. The p-value was indicated in the figures as follows: * p-value$<$0.05, ** p-value$<$0.01, *** p-value$<$0.001, **** p-value$<$0.0001, ns non-significant.

## Results

Both CPC [12,13,19] and CHX [23] have been previously shown to have antiviral properties. We first sought to confirm the efficacy of these antimicrobial agents against two of the most prevalent respiratory viruses, IAV and RSV. IAV (IAV/WSN/33) was incubated with CPC or CHX, at concentrations commonly found in pharmaceutical products, for approximately 2 minutes. Next, the virus was diluted and used to infect MDCK cells. After 48 hours of infection, the viral load was assessed by $TCID_{50}$. A schematic representation of the experimental setup can be found in Fig 1. In these experiments, we used SDS at 0.05% as a positive control and PBS solution as a negative control. Additionally, a sample was mock-treated and the resulting viral load was used to normalize the experimental values. Our results indicate that CPC can reduce 90% and 99.9% of IAV infectivity at 0.05 and 0.1% respectively. On the other hand, CHX reduced ~ 90% of IAV viability at any of the tested concentrations (0.25, 0.12, and 0.06%, Fig 2A). Shorter incubation times were tested, and no differences were found (S1 Fig).

Next, we tested the effect of CPC and CHX on RSV (Fig 2B). As with IAV, RSV was treated at the indicated concentration for approximately 2 minutes. Next, the virus, and the

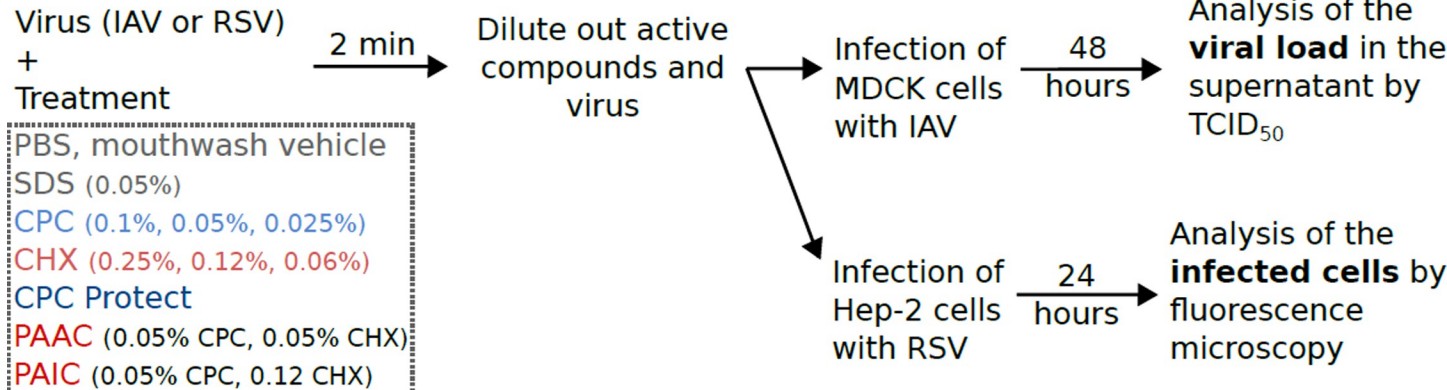

**Fig 1. Schematic representation of the experimental setup.** IAV (IAV/WSN/33) or RSV (RSV-mKate) were incubated with CPC or CHX for approximately 2 minutes at the indicated concentrations. Alternatively, the virus was treated with the commercial formulations CPC Protect (0.07% CPC), Perio Aid Active Control (PAAC, 0.05% CPC and 0.05% CHX), and Perio Aid Intensive Care (PAIC, 0.05% CPC and 0.12% CHX). We also included treatments with PBS and mouthwash vehicle as a negative control, SDS at the indicated concentration as a positive control, and mock treatment to calculate the maximum infection efficiency. Next, the viruses were diluted and used to infect MDCK or HeLa (Hep-2) cells. After 48 (IAV) or 24 hours (RSV) the viral load was measured by TCID$_{50}$ (IAV) or the number of cells infected counted by fluorescence microscopy (RSV).

compounds, were diluted, and the resulting dilution was used to infect HeLa-derived Hep-2 cells (Fig 1). We used an RSV strain that expresses the mKate2 protein [28] to facilitate the identification of infected cells at 24 hours post-infection. Our results indicate that CPC is capable of blocking RSV infection, ~ 99.9% reduction in infected cells at concentrations of 0.025% or higher. CHX impeded 90 or 99.9% of the RSV infection at 0.12 and 0.25% respectively.

To explore the antiviral properties of mouthwash solutions containing CPC and or CHX, several formulations for oral care were prepared in the laboratory of the DENTAID Research Center. Table 1 shows the full list of ingredients of the formulations. Three distinct formulations were evaluated: Vitis CPC Protect, containing 0.07% CPC, an oral rinse recommended for daily use to prevent and reduce dental plaque formation (CPC Protect), Perio Aid Active control containing 0.05% CPC together with 0.05% CHX recommended for periodontitis control and patients recovering from dental surgery (PAAC) and Perio Aid Intensive Care 0.05% CPC, 0.12% CHX intended for limited-term use as a coadjuvant for patients undergoing periodontal treatment and/or surgery in the oral cavity (PAIC). As a control, we included PBS and mouthwash vehicle with no CPC or CHX prepared by the DENTAID Research Center. As in previous experiments, IAV or RSV-mKate2 were incubated with the mentioned mouthwash formulations for 2 minutes. Next, the solutions were diluted and used to infect MDCK or Hep-2 cells respectively. In the case of IAV, the viral load was measured 48 hours post-infection by TCID$_{50}$. RSV infectivity was measured by fluorescent microscopy at 24 hours post-infection. Our results indicate that all three formulations reduced IAV infectivity below 99.9% compared to the PBS-treated samples. On the other hand, RSV viability was reduced by 90% with PAAC and below 99.9% with Vitis CPC Protect and PAIC in comparison with the PBS-treated samples (Fig 2C).

Finally, we wanted to ensure that the antiviral effect seen for CPC or CHX as stand-alone treatments or as components of mouthwash formulations was not the result of cellular toxicity. To do so, we incubated CPC, CHX, Vitis CPC Protect, PAAC, or PAIC at multiple concentrations with MDCK and HeLa cells (Fig 3) and after 24 hours we measured the cell viability. We included PBS and mouthwash vehicle as controls. None of the treatments showed toxicity at the concentrations applied to the cells. Nonetheless, some cellular toxicity was observed at elevated concentrations both CPC and CHX. These results confirm that the antiviral properties

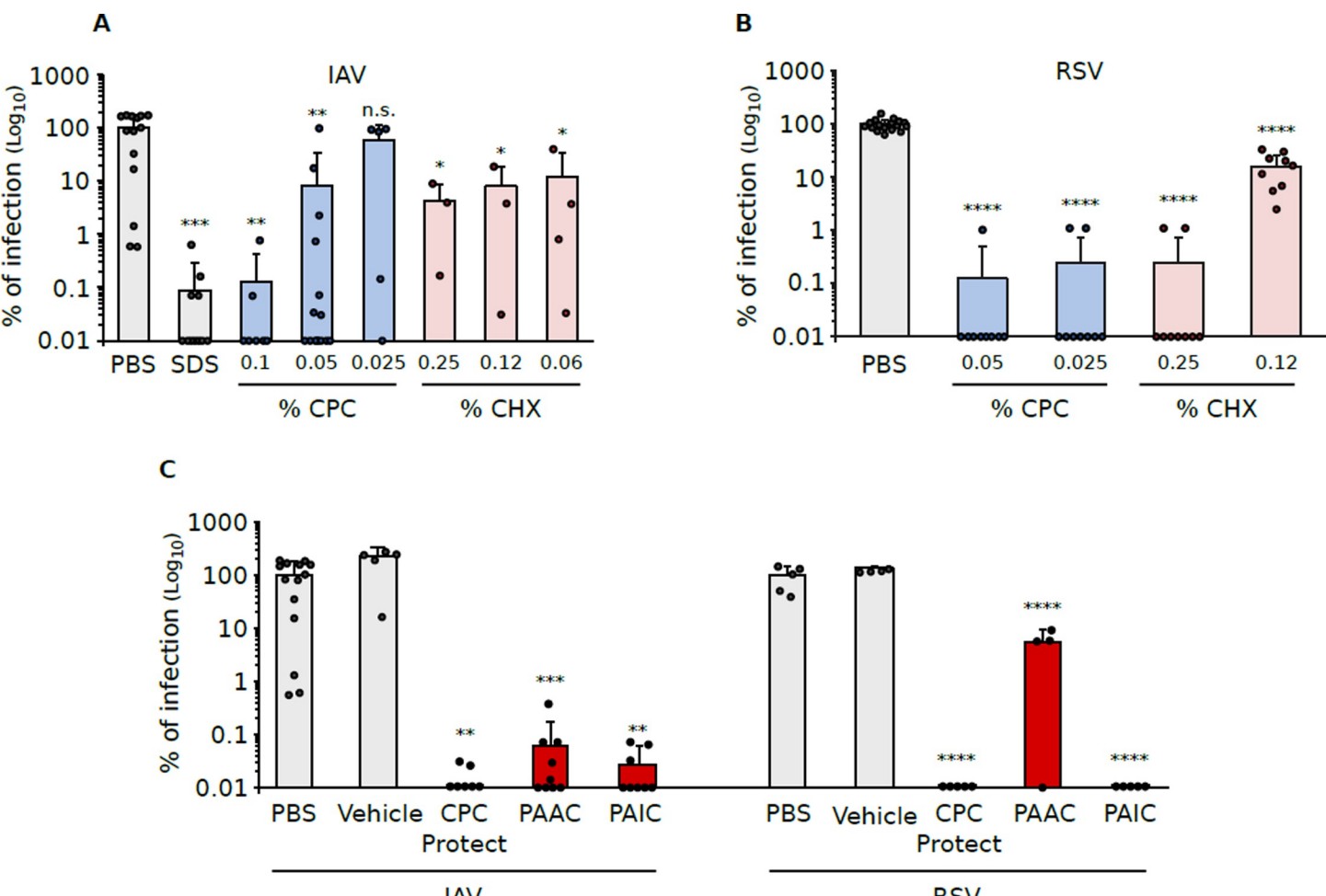

**Fig 2. Antiviral activity of CPC and/or CHX-containing solutions. A**. IAV was treated with CPC or CHX at different concentrations. The treated virus was then used to infect MDCK cells. After 48 hours the virus in the supernatant was collected and tittered by $TCID_{50}$. As control treatments, we used PBS or SDS at 0.05%. **B**. RSV-mKate was treated with CPC or CHX at different concentrations. The treated virus was then used to infect Hep-2 cells. After 24 hours the number of infected cells was quantified as an indication of infective viral particles. In A and B we use PBS or SDS at 0.05% as control treatments. **C**. Treatment of IAV or RSV with mouthwash formulations. Treatment and the subsequent infections were done as previously described. Vitis CPC Protect (CPC Protect) contains 0.07% CPC. Perio Aid Active control (PAAC) contains 0.05% CPC and 0.05% CHX. Perio Aid Intensive Care (PAIC) contains 0.05% CPC and 0.12% CHX. In all panels, the graphs show the average and standard deviation of at least three experiments. The individual value of each experiment is represented by a dot. The level of significance (p-value, two-tailed homoscedastic t-test) when comparing the PBS treatment (gray bar) vs. any of the treatments is shown above the bars; $H_0$: μ = average viral titer of PBS treated samples. * p-value<0.05, ** p-value<0.01, ***p-value<0.001, ****p-value<0.0001, ns non-significant. As an additional control, we included the mouthwash formations vehicle (vehicle).

of CPC and CHX-containing mouthwash solutions do not stem from toxicity. Note that, similar results were obtained for HeLa-derived Hep-2 cells, for which confluency was not affected by the diluted solutions used to infect the cells.

## Discussion

Numerous respiratory viruses utilize the oral cavity as the portal of entry, a site of replication, and more importantly, an exit route. With the ultimate goal of identifying additional measures to limit respiratory viral infection and transmission, we have tested whether CPC and or CHX can decrease the infectivity of two of the most medically relevant respiratory viruses, that is, IAV and RSV. Our results indicate that CPC and CHX can decrease IAV and RSV infectivity *in vitro*. Furthermore, our data indicates that commercial mouthwash preparations containing

**Table 1. Mouthwash composition (INCI lists).**

| Mouthwash | Ingredients |
|---|---|
| **CPC Protect** | Aqua, Glycerin, Propylene, Glycol, Xylitol, Poloxamer 407, Methylparaben, Cetylpyridium Chloride (0.07%), Propylparaben, Menthone Glycerin Acetal, Sodium Saccharin, Aroma, C.I. 42051 |
| **PAAC** | Aqua, Glycerin, Propylene Glycol, Xylitol, PEG-40 Hydrogenated Castor Oil, Methylparaben, Chlorhexidine Digluconate (0.05%), Cetylpyridinium Chloride, Potassium Acesulfame, Ethyparaben, Sodium Saccharin, Neohesperidin Dihydrochalcone, Aroma, C.I. 42051 |
| **PAIC** | Aqua, Glycerin, Propylene Glycol, Xylitol, Chlorhexidine Digluconate (0.12%), PEG-40 Hydrogenated Castor Oil, Cetylpyridinium Chloride (0.05%), Potassium Acesulfame, Sodium Saccharin, Neohespiridin Dichalcone, Aroma, C.I. 42090 |
| **Vehicle** | Aqua, Glycerin, Propylene, Glycol, Xylitol, Poloxamer 407, Methylparaben, Cetylpyridium Chloride (0.07%), Propylparaben, Menthone Glycerin Acetal, Sodium Saccharin, Aroma, C.I. 42051 |

Note that we incubated the viruses with the mouthwash solution at a 1:1 ratio. Therefore, the working concentration of CPC and CHX are 0.035% CPC in Vitis CPC Protect, 0.025% CPC, 0.025% CHX in PAAC, and 0.025% CPC, 0.06% CHX in PAIC.

either CPC alone or CPC and CHX combined were effective in decreasing the viral viability of these two respiratory viruses, once again *in vitro*. Conclusions were based on a statistical analysis where the level of significance when comparing the PBS treatment vs. any of the treatments was calculated using a two-tailed homoscedastic t-test ($H_0$: $\mu$ = average viral titer of PBS treated samples).

We devoted our study to CPC and CHX because they are two of the most common antimicrobial agents found in pharmaceutical compounds. The range of concentrations for these two compounds was selected based on the product information of the main over-the-counter mouthwashes according to their active ingredients list, a summary of this information can be found in Carrouel et al. [9].

CPC and other quaternary ammonium compounds have been used for decades against a variety of pathogens [10,11]. Likewise, CHX has been used extensively as a disinfectant and antiseptic and is commonly used by the pharmaceutical and cosmetic industry. However, we should remember that CPC and CHX are not the only antiviral agents included in the spectrum of commercial mouthwashes and toothpaste [9]. On the other hand, we have focused our efforts on IAV and RSV due to the sparse literature on the effect of CPC and CHX on these two viruses, despite their medical importance, compared with other respiratory viruses, particularly SARS-CoV-2 [9,29,30].

The most comprehensive study up to date on the effect of CPC on IAV was conducted by Popkin et al. [19]. In this assay, the authors evaluated the effect of CPC as a treatment for IAV both *in vitro* and *in vivo*. According to their objective, the authors used longer incubation times and lower CPC concentrations compared to our experimental design. Nonetheless, the authors observed that CPC possesses virucidal activity against multiple strains of influenza virus by targeting and disrupting the viral envelope. Comparatively, there is more literature on the effect of CHX on IAV [24,31,32]. In all these reports, despite the experimental differences, a reduction in IAV titers was observed after CHX treatment.

However, there is little literature on the effect of CHX on RSV infectivity [33]. Contrary to our results, Platt et al. did not observe a virucidal effect of CHX on RSV. No reports were found on CPC and RSV. All in all, our results confirm the antiviral properties of CPC and CHX against IAV and include RSV as a susceptible microorganism to these two compounds. Given these results, the data available in the literature [5,8,9,11,13–15,19–21,34], and the mechanisms of action of both CPC and CHX [10,22,35–38] we could expect similar results for other enveloped respiratory viruses.

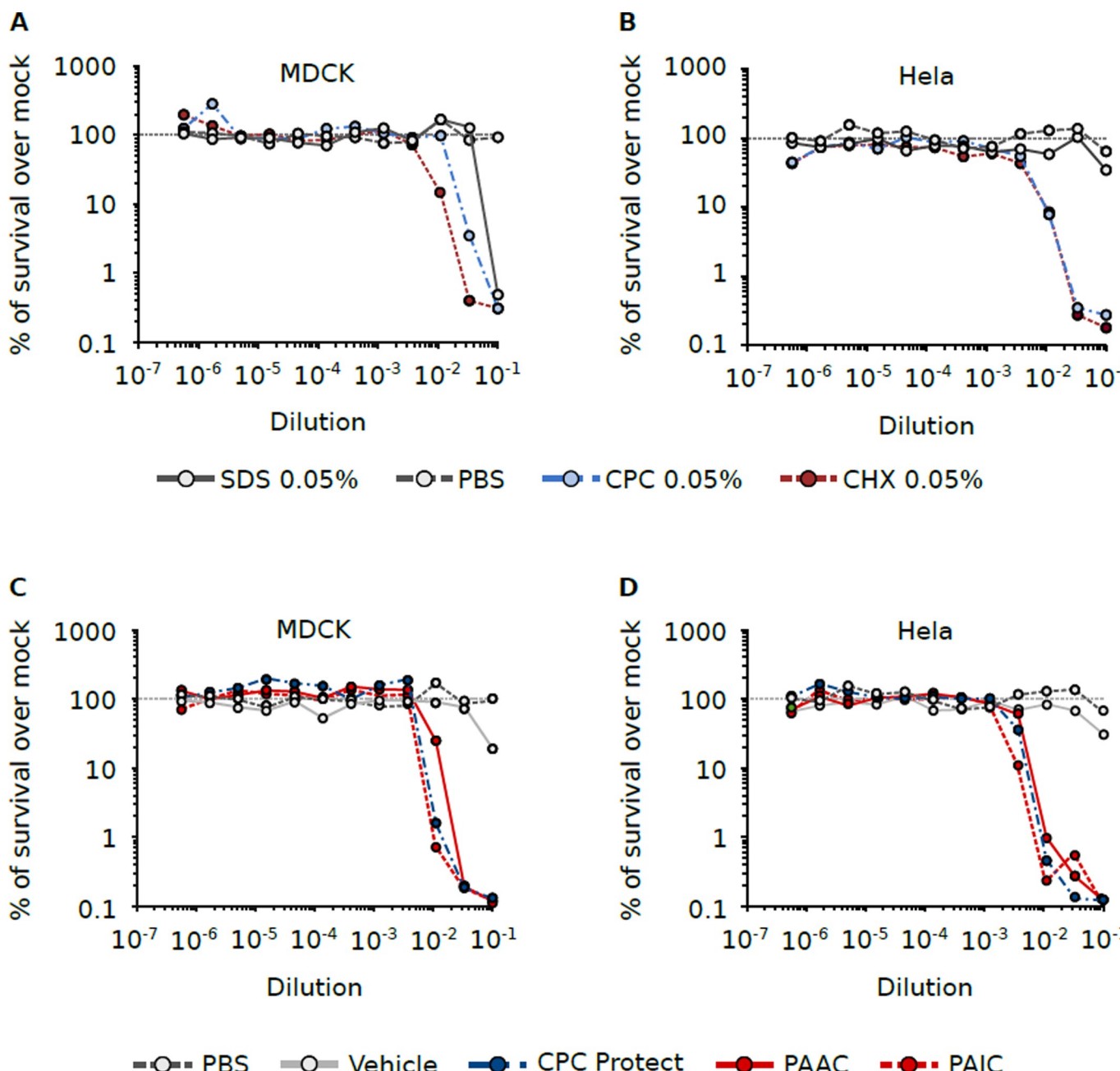

**Fig 3. Cytotoxicity of CPC and CHX-containing solutions.** HeLa and MDCK cells were incubated with CPC or CHX at different concentrations for 24 hrs. Next, the cell viability was measured using Promega's Cell Titer Glo. As control treatments, we used PBS and SDS. Compounds were serially diluted in PBS (x-axes) from the stock solutions (SDS 0.05%, CPC 0.05%, and CHX 0.05%). The toxicity of mouthwash solutions was also tested. Vitis CPC Protect (CPC Protect) contains 0.07% CPC. Perio Aid Active control (PAAC) contains 0.05% CPC and 0.05% CHX. Perio Aid Intensive Care (PAIC) contains 0.05% CPC and 0.12% CHX. Each of the mouthwash solutions was diluted in PBS (dilution indicated in the x-axes) before cell treatment. In this case, we included treatment with a mouthwash vehicle as an additional control. The y-axes indicated the percentage of survival compared with the average value for mock-treated cells (horizontal dashed line). Each point represents the average of at least three independent experiments.

Structurally IAV and RSV are similar [39]; both are RNA viruses with a matrix protein coat and a lipid envelope. Accordingly, we see similar results with both viruses. However, a detailed analysis indicates that RSV is more sensitive to both CPC and CHX. Nonetheless, these differences might be the result of variations in the experimental setup. We measured the IAV viral

load in the supernatant after 48 hours of infection by standard $TCID_{50}$. To analyze RSV viability, we counted infected cells at 24 hours post-infection by monitoring the mKate2 fluorescence. These differences might also account for the larger variation within replicas observed with IAV.

Before we can claim that the use of a mouthwash containing CPC or CHX alone or in combination might represent an effective measure to limit infection or spread of enveloped respiratory viruses infecting the oral cavity further studies must be conducted. First, the reduction in the viral load after the use of an adequate mouthwash should be demonstrated *in vivo*. Some effort has been made in this direction, particularly in the case of SARS-CoV-2. However, given the many technical and logistical difficulties associated with this type of assay, the results remain inconclusive [29]. Next, the time window after the use of the mouthwash where the viral load is affected should be established. Finally, if the viral load is sufficiently reduced and the reduction lasts a significant period a randomized clinical trial should be conducted to test the transmission of respiratory viruses.

## Supporting information

**S1 Fig. Time of exposure.** We tested the effect of the exposure time to CPC. To do so, IAV/WSN/33 was incubated with CPC at 0.1% for 2 minutes, 1 minute, or 30 seconds. Next, the virus was diluted and used to infect MDCK cells as previously described. After 48 hours of infection, the viral load was assessed by TCID50. We used a 2-minute treatment with SDS at 0.05% as a positive control and PBS solution as a negative control. No differences in the viral load were observed between the 2 minutes, 1 minute, or 30 seconds exposure. (PDF)

## Acknowledgments

We thank P. Selvi for her excellent technical assistance.

## Author Contributions

**Conceptualization:** Maria Jesús García-Múrria, Ron Geller, Ismael Mingarro, Luis Martinez-Gil.

**Data curation:** Luis Martinez-Gil.

**Formal analysis:** Luis Martinez-Gil.

**Investigation:** Marina Rius-Salvador, Maria Jesús García-Múrria, Luciana Rusu, Luis Martinez-Gil.

**Project administration:** Maria Jesús García-Múrria, Ismael Mingarro, Luis Martinez-Gil.

**Resources:** Manuel Bañó-Polo, Rubén León, Ron Geller, Ismael Mingarro.

**Supervision:** Ismael Mingarro, Luis Martinez-Gil.

**Validation:** Ron Geller.

**Writing – original draft:** Luis Martinez-Gil.

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
