## [Decision Letter · Decision Letter 0]

5 Nov 2023

PONE-D-23-32284Cetylpyridinium chloride and chlorhexidine show antiviral activity against Influenza A virus and Respiratory Syncytial virusPLOS ONE

Dear Dr. Martinez-Gil,

Thank you for submitting your manuscript to PLOS ONE. After careful consideration, we feel that it has merit but does not fully meet PLOS ONE’s publication criteria as it currently stands. Therefore, we invite you to submit a revised version of the manuscript that addresses the points raised during the review process.

We look forward to receiving your revised manuscript.

Kind regards,

Ajaya Bhattarai

Academic Editor

PLOS ONE

Journal Requirements:

"DENTAID partially funded this study and supplied mouthwash formulations. This work was supported by the Generalitat Valenciana (PROMETEO/2022/062) and grant PID2020-119111GB-I00 by MCIN/AEI/10.13039/501100011033. R.G. acknowledges funding from the European Commission – NextGenerationEU (Regulation EU 2020/2094), through the CSIC's Global Health Platform (PTI Salud Global). M.R-S is the recipient of a predoctoral contract from the Spanish Ministry of Science and Innovation (PRE2021-101042 by MCIN/AEI/10.13039/501100011033)."

3. Please expand the acronym “MCIN/AEI” (as indicated in your financial disclosure) so that it states the name of your funders in full.

"DENTAID partially funded this study and supplied mouthwash formulations. This work was supported by the Generalitat Valenciana (PROMETEO/2022/062) and grant PID2020-119111GB-I00 by MCIN/AEI/10.13039/501100011033. R.G. acknowledges funding from the European Commission – NextGenerationEU (Regulation EU 2020/2094), through the CSIC's Global Health Platform (PTI Salud Global). M.R-S is the recipient of a predoctoral contract from the Spanish Ministry of Science and Innovation (PRE2021-101042 by MCIN/AEI/10.13039/501100011033). We thank P. Selvi for excellent technical assistance."

"DENTAID partially funded this study and supplied mouthwash formulations. This work was supported by the Generalitat Valenciana (PROMETEO/2022/062) and grant PID2020-119111GB-I00 by MCIN/AEI/10.13039/501100011033. R.G. acknowledges funding from the European Commission – NextGenerationEU (Regulation EU 2020/2094), through the CSIC's Global Health Platform (PTI Salud Global). M.R-S is the recipient of a predoctoral contract from the Spanish Ministry of Science and Innovation (PRE2021-101042 by MCIN/AEI/10.13039/501100011033)."

Reviewers' comments:

Reviewer's Responses to Questions

**Comments to the Author**

1. Is the manuscript technically sound, and do the data support the conclusions?

Reviewer #1: Yes

Reviewer #2: Partly

Reviewer #3: Partly

2. Has the statistical analysis been performed appropriately and rigorously? 

Reviewer #1: Yes

Reviewer #2: N/A

Reviewer #3: No

3. Have the authors made all data underlying the findings in their manuscript fully available?

Reviewer #1: Yes

Reviewer #2: Yes

Reviewer #3: Yes

4. Is the manuscript presented in an intelligible fashion and written in standard English?

Reviewer #1: Yes

Reviewer #2: Yes

Reviewer #3: Yes

5. Review Comments to the Author

Reviewer #1: The present in vitro study investigated the antiviral efficacy of cetylpyridinium chloride and chlorhexidine digluconate and various combinations of both for inactivation of Influenza A virus and Respiratory Syncytial virus. The topic of the study is timely, the study has been well-designed and performed diligently. Therefore, there are just some minor points needing revision, as follows:

Title/Abstract:

I would suggest to include the information that the present study is an in vitro investigation in the title and abstract

Introduction:

- ln. 57: please add the following paper as reference for the effects of a combination of CHX and CPC toward SARS-CoV-2: https://pubmed.ncbi.nlm.nih.gov/36942423/

- ln. 61-68: please include references for the statements regarding CHX, e.g. https://pubmed.ncbi.nlm.nih.gov/30967854/

Methods:

- I would suggest to include a table with all test compounds, the respective concentrations of antiseptics and the other ingredients (INCI lists)

Results & Discussion:

- please include the antiseptic concentrations for the commercial products in Figure 1

- it is a bit confusing to name the two products PerioAid1 and PerioAid2. What about PAAC for Perio Aid Active control and PAIC for Perio Aid Intensive Care?

- please also add information on the ingredients of the mouthwash vehicle (see comment above, suggestion of adding a table)

- please include an outlook for future studies, e.g. to conduct a clinical trial and test the effects on intraoral viral load and infectivity (cf.: https://pubmed.ncbi.nlm.nih.gov/35897159/ )

Reviewer #2: Dear authors,

Your article untitled “Cetylpyridinium chloride and chlorhexidine antiviral activity against IAV and RSV” is an interesting article. However, it should be entirely revised taking into account my comments:

Title

1. Please as your study is based on in vivo studies, you should consider to include “in vitro” in your tittle.

Résumé

1. The summary is very short. The material and method should be explained.

2. It could be interesting to structure your abstract: Background, Mat met, Results and then conclusion

Introduction

1. L68 For CHX it will be interesting to precise, as you did for Cetylpyridium, the virus concerned by the antiviral activity. Indeed, CHX acts against CMV, FluV, HBV, SARS-CoV2… (DOI: 10.1177/0022034520967933)

2. L70-73 Could you formulate the aim of your study?

Materials and Methods

1. Could you precise for each material (MDCK, DMEM…) the manufacturer, the city and the country?

2. Why did you dilute CPC in PBS and CHX in water? Why did not use the same solution: water or PBS?

3. Why did you do different dilutions for CPC and CHX?

4. L85 to 94 Please can you give more explanations? The steps are not clear for me. You write “the solutions” (line 88): What are these solutions?

5. How did you choose the commercial mouthwashes? Did you test all the mouthwashes commercially available and containing CPC ou CHX?

Results and discussion

1. This section should be entirely revised because there is only results and no discussion. Discussion comparing the results of this study with previous ones is essential.

2. Please add in the discussion section the limitations of this study

3. Please discuss your choice of analyzing only CPC and CHX because other products could be interesting (DOI: 10.1177/0022034520967933)

References

1. It is important to include more references to discuss your results

1. Line 178 Replace “Acknoledments” by “Acknowledgments”

Reviewer #3: Dear authors,

thank you for the opportunity to review the interesting manuscript dealing with a clinically relevant topic.

However, there are some major point contradicting publication, as listed below:

Title:

From the title, the reader gets no information about the kind of study (is it a clinical study, is it an in vitro study etc.). Please adapt.

Abstract:

The abstract does neither contain information about the methods used to obtain the findings, nor the statsitical methods, nor about numercial data. Basically, the reader gets no information about the study at all from the abstract.

The following sentence is unclear and not precise, please provide more information: "Not only do new viruses

39 enter the playground but the impact of old ones is also augmented due to climate change(4)."

There is another clinical study with CHX/CPC combination against SARS-CoV2. Please add

https://pubmed.ncbi.nlm.nih.gov/36942423/

Regarding the mechanisms, you mostly described antibacterial effects, but almost no antiviral mechanisms. As viruses are within the scope of your study, you should add these informations.

Please use a clearer (null)-hypothesis for your study.

Mat/Met:

Please provide a definition for each abbreviation with the first appearance (e.g. MDCK).

Why did you use different concentrations fpr CHX and CPC? This should be explained in the Mat/Met or the discussion part.

It is not clear, if you used experimental or commercially available solutions of CHX or CPC. Please rewrite the corresponding part.

Please explain the RSV (with abbreviation definition as well) production in more detail.

Please add a part regarding the ethical disclosure for using cells and a paragraph explaining the statistical methods.

Results/Discussion:

These parts should be seperated according to the submission guidelines. Furthermore, some informations are missing:

-Which statistical tests are the base of the asterisks and statistcial differences?

-What probable explanation do you have for the different performances of CPC and CHX and whiy did you use different concentrations from both substances?

-Please add a paragraph diuscussing the limitations of the in vitro design, especially regarding differences to the oral cavity.

6. PLOS authors have the option to publish the peer review history of their article (what does this mean?). If published, this will include your full peer review and any attached files.

Reviewer #1: No

Reviewer #2: No

Reviewer #3: No

---

## [Author Response · Author response to Decision Letter 0]

28 Nov 2023

Response to reviewers

We thank the reviewers for their helpful feedback and for their suggestions to improve our manuscript. We hope we have fulfilled the reviewers’ concerns sufficiently. As requested, below you’ll find a point-by-point response to each comment. If any additional information is necessary, please do not hesitate to contact us.

Changes in the Manuscript have been highlighted in red. 

Reviewer #1

The present in vitro study investigated the antiviral efficacy of cetylpyridinium chloride and chlorhexidine digluconate and various combinations of both for inactivation of Influenza A virus and Respiratory Syncytial virus. The topic of the study is timely, the study has been well-designed and performed diligently. Therefore, there are just some minor points needing revision, as follows:

Title/Abstract:

I would suggest to include the information that the present study is an in vitro investigation in the title and abstract

We have changed the manuscript title as requested. 

Introduction:

- ln. 57: please add the following paper as reference for the effects of a combination of CHX and CPC toward SARS-CoV-2: https://pubmed.ncbi.nlm.nih.gov/36942423/

- ln. 61-68: please include references for the statements regarding CHX, e.g. https://pubmed.ncbi.nlm.nih.gov/30967854/

Both references have been included in the manuscript. 

Methods:

- I would suggest to include a table with all test compounds, the respective concentrations of antiseptics and the other ingredients (INCI lists)

We have included a table, following a standard INCI List format (Table 1), where all the ingredients of the commercial mouthwash formulations utilized in this work were listed. 

Results & Discussion:

- please include the antiseptic concentrations for the commercial products in Figure 1

We have modified Figure 1 following the reviewer recommendation. 

- it is a bit confusing to name the two products PerioAid1 and PerioAid2. What about PAAC for Perio Aid Active control and PAIC for Perio Aid Intensive Care?

We thank the reviewer for the suggestion. It is indeed much clearer with the suggested nomenclature. The text and the figures have been modified accordingly. 

- please also add information on the ingredients of the mouthwash vehicle (see comment above, suggestion of adding a table)

The requested information has been included in Table 1.

- please include an outlook for future studies, e.g. to conduct a clinical trial and test the effects on intraoral viral load and infectivity (cf.: https://pubmed.ncbi.nlm.nih.gov/35897159/ )

We have included these thoughts in the new Discussion section. 

Reviewer #2

Dear authors,

Your article untitled “Cetylpyridinium chloride and chlorhexidine antiviral activity against IAV and RSV” is an interesting article. However, it should be entirely revised taking into account my comments:

Title

1. Please as your study is based on in vivo studies, you should consider to include “in vitro” in your tittle.

The title has been revised. It now reads: Cetylpyridinium chloride and chlorhexidine show antiviral activity against Influenza A virus and Respiratory Syncytial virus in vitro

Résumé

1. The summary is very short. The material and method should be explained.

2. It could be interesting to structure your abstract: Background, Mat met, Results and then conclusion

Following the reviewer’s recommendation, we have expanded the Abstract section. We have also divided it into Background, M&M, Results and, Conclusion sub-sections.

Introduction

1. L68 For CHX it will be interesting to precise, as you did for Cetylpyridium, the virus concerned by the antiviral activity. Indeed, CHX acts against CMV, FluV, HBV, SARS-CoV2… (DOI: 10.1177/0022034520967933)

We have extended the introduction section to include the requested information. We have also included new references. 

2. L70-73 Could you formulate the aim of your study?

We apologize if the aims of the study were not clear. The corresponding section of the introduction has been rewritten. 

Materials and Methods

1. Could you precise for each material (MDCK, DMEM…) the manufacturer, the city and the country?

The missing information has been included. 

2. Why did you dilute CPC in PBS and CHX in water? Why did not use the same solution: water or PBS?

3. Why did you do different dilutions for CPC and CHX?

4. L85 to 94 Please can you give more explanations? The steps are not clear for me. You write “the solutions” (line 88): What are these solutions?

We deeply apologize for the confusing explanation. We have rewritten the entire paragraph and hope it is now clear; It was not our intention to mislead the reader. 

5. How did you choose the commercial mouthwashes? Did you test all the mouthwashes commercially available and containing CPC ou CHX?

As stated in the manuscript DENTAID partially funded this study and supplied the mouthwashes. We believe the commercial formulations used in our work provide a solid indication of what we can expect from other mouthwashes with similar CPC and CHX concentrations. 

Results and discussion

1. This section should be entirely revised because there is only results and no discussion. Discussion comparing the results of this study with previous ones is essential.

2. Please add in the discussion section the limitations of this study

3. Please discuss your choice of analyzing only CPC and CHX because other products could be interesting (DOI: 10.1177/0022034520967933)

These and other thoughts are now discussed in the newly added Discussion section.

References

1. It is important to include more references to discuss your results

As a result of extending the introduction and particularly the discussion section, we have included new references. 

1. Line 178 Replace “Acknoledments” by “Acknowledgments”

We thank the reviewer for identifying this typo, it has been corrected. 

Reviewer #3

Dear authors,

Thank you for the opportunity to review the interesting manuscript dealing with a clinically relevant topic. However, there are some major point contradicting publication, as listed below:

Title:

From the title, the reader gets no information about the kind of study (is it a clinical study, is it an in vitro study etc.). Please adapt.

We have changed the title to indicate the type of study conducted.

Abstract:

The abstract does neither contain information about the methods used to obtain the findings, nor the statsitical methods, nor about numercial data. Basically, the reader gets no information about the study at all from the abstract.

We have rewritten the abstract. It is now divided into Background, M&M, Results and, Conclusion sub-sections. We hope the new wording and structure provide enough information about the content of the manuscript. 

The following sentence is unclear and not precise, please provide more information: "Not only do new viruses enter the playground but the impact of old ones is also augmented due to climate change(4)."

We wanted to indicate that not only new respiratory viruses are emerging but also that the impact of current circulating viruses is increasing due to climate change. We have rewritten the sentence. We hope it is clearer now. 

There is another clinical study with CHX/CPC combination against SARS-CoV2. Please add

https://pubmed.ncbi.nlm.nih.gov/36942423/

As suggested by the reviewer we have included the missing reference. 

Regarding the mechanisms, you mostly described antibacterial effects, but almost no antiviral mechanisms. As viruses are within the scope of your study, you should add these informations.

When describing the mechanism of action we referred to “biological membranes” which include bacterial membranes but also the viral membrane in the case of enveloped viruses. We modify the text to clarify this concept. 

Please use a clearer (null)-hypothesis for your study.

We have included in Figure legend 2 the null hypothesis of our statistical analysis: “H0: μ=average viral titer of PBS treated samples.”

Mat/Met:

Please provide a definition for each abbreviation with the first appearance (e.g. MDCK).

We apologize for the missing information. We have included the definition for each of the abbreviations. 

Why did you use different concentrations for CHX and CPC? This should be explained in the Mat/Met or the discussion part. 

The range of concentrations for CPC and CHX was selected based on the product information of the main over-the-counter mouthwashes according to their active ingredients list. This explanation has been included in the Discussion section. 

It is not clear, if you used experimental or commercially available solutions of CHX or CPC. Please rewrite the corresponding part.

The CPC, CHX, and mouthwash solutions were provided by DENTAID. This information has been included in the Material and Methods. 

Please explain the RSV (with abbreviation definition as well) production in more detail.

We have incorporated a section that indicates how the RSV-mKate2 was generated. Furthermore, we have extended the information regarding the RSV viability assays. 

Please add a part regarding the ethical disclosure for using cells and a paragraph explaining the statistical methods.

As far as we are concerned PLoS ONE does not requires an ethical statement for the use of cell cultures. Nonetheless, we have informed the editorial team of the reviewer’s concern and provided them with an ethical statement in case they choose to include it. 

We have included in the M&M a section regarding the Statistical analysis. Note that we have kept some of this information in the Figure 2 legend. 

Results/Discussion:

These parts should be seperated according to the submission guidelines. Furthermore, some informations are missing:

-Which statistical tests are the base of the asterisks and statistcial differences?

-What probable explanation do you have for the different performances of CPC and CHX and whiy did you use different concentrations from both substances?

-Please add a paragraph diuscussing the limitations of the in vitro design, especially regarding differences to the oral cavity.

As requested we have separated the Results and Discussion sections. We apologize for the condensed version of the original manuscript. All the requested information is included in the new Discussion section.

---

## [Decision Letter · Decision Letter 1]

6 Dec 2023

PONE-D-23-32284R1Cetylpyridinium chloride and chlorhexidine show antiviral activity against Influenza A virus and Respiratory Syncytial virus in vitroPLOS ONE

Dear Dr.  Martinez-Gil,

Thank you for submitting your manuscript to PLOS ONE. After careful consideration, we feel that it has merit but does not fully meet PLOS ONE’s publication criteria as it currently stands. Therefore, we invite you to submit a revised version of the manuscript that addresses the points raised during the review process. Please submit your revised manuscript by Jan 20 2024 11:59PM. If you will need more time than this to complete your revisions, please reply to this message or contact the journal office at plosone@plos.org. Please include the following items when submitting your revised manuscript:A rebuttal letter that responds to each point raised by the academic editor and reviewer(s). You should upload this letter as a separate file labeled 'Response to Reviewers'.A marked-up copy of your manuscript that highlights changes made to the original version. You should upload this as a separate file labeled 'Revised Manuscript with Track Changes'.An unmarked version of your revised paper without tracked changes. You should upload this as a separate file labeled 'Manuscript'.

We look forward to receiving your revised manuscript.

Kind regards,

Ajaya Bhattarai

Academic Editor

PLOS ONE

Additional Editor Comments:

The academic editor suggests major revisions.

Reviewers' comments:

Reviewer's Responses to Questions

**Comments to the Author**

1. If the authors have adequately addressed your comments raised in a previous round of review and you feel that this manuscript is now acceptable for publication, you may indicate that here to bypass the “Comments to the Author” section, enter your conflict of interest statement in the “Confidential to Editor” section, and submit your "Accept" recommendation.

Reviewer #1: All comments have been addressed

Reviewer #3: (No Response)

2. Is the manuscript technically sound, and do the data support the conclusions?

Reviewer #1: Yes

Reviewer #3: Yes

3. Has the statistical analysis been performed appropriately and rigorously? 

Reviewer #1: Yes

Reviewer #3: Yes

4. Have the authors made all data underlying the findings in their manuscript fully available?

Reviewer #1: Yes

Reviewer #3: Yes

5. Is the manuscript presented in an intelligible fashion and written in standard English?

Reviewer #1: Yes

Reviewer #3: Yes

6. Review Comments to the Author

Reviewer #1: All comments raised by this reviewer have been addressed sufficiently. Congrats to a very nice paper!

Reviewer #3: Dear authors,

thank you for the renewed opportunity to review the manuscript.

Although improved in some points, m there are still major points that need adaptation, especially in the newly written parts.

Abstract:

The results part, although in better structure now, does still not provide numerical data. Please add.

Typo: "CPD" in the conclusion of the abstract.

Introduction:

Please be more concise regarding the mechanism of climate change leading to more viruses.

Mat/Met:

As "pfu" is not an SI-unit, you should define it with the first appearance.

The bracket containing DMEM and FBS is unclear.

As for all other companies, the location and state of Dentaid should be named with the first appearance.

This sentence is extremely complicated and hard to understand and should therefore be simplified and shortened: "To generate this virus the RSV infectious clone system, obtained from BEI Resources (NIAID, NIH: Bacterial artificial chromosome plasmid pSynkRSV-I19F containing antigenomic cDNA from RSV A2-Line19F, NR-3646), was transfected into HEK-293 cells together with a codon-optimized T7 polymerase (Addgene, United States, 65974), and an RSV N, P, M2-1, and L expression plasmids at a ratio of 4:2:2:2:1."

Discussion:

L.287: How do you define a "large study"? Please specify.

Some of the new paragraphs in the discussion are lacking relevant references, especially 295-308. Please add.

7. PLOS authors have the option to publish the peer review history of their article (what does this mean?). If published, this will include your full peer review and any attached files.

Reviewer #1: No

Reviewer #3: No

---

## [Author Response · Author response to Decision Letter 1]

12 Dec 2023

Reviewers' comments

We thank the reviewers for their compliments. We hope we have fulfilled the reviewers’ concerns sufficiently. As requested, below you’ll find a point-by-point response to each comment. If any additional information is necessary, please do not hesitate to contact us.

Responses are provided in blue below each comment in the Reviewers' comments document. Changes in the Manuscript have been highlighted in red. 

Reviewer #1: All comments raised by this reviewer have been addressed sufficiently. Congrats to a very nice paper!

Thank you very much. We truly believe the reviewers have improve the quality of the manuscript.

Reviewer #3: Dear authors,

Thank you for the renewed opportunity to review the manuscript.

Although improved in some points, there are still major points that need adaptation, especially in the newly written parts.

Thank you very much for your time and dedication to improving the quality of our manuscript.

Abstract:

The results part, although in better structure now, does still not provide numerical data. Please add.

We have incorporated the reduction values associated with each of the tested treatments for both RSV and IAV. 

Typo: "CPD" in the conclusion of the abstract.

Thanks for identifying this typo. We have fixed this error and checked the entire document for additional typos. 

Introduction:

Please be more concise regarding the mechanism of climate change leading to more viruses.

We have included more information regarding the relation between climate change and the increase in respiratory infections. However, we have kept it brief since this is but a motivation to focus on respiratory viruses. 

Mat/Met:

As "pfu" is not an SI-unit, you should define it with the first appearance.

The pfu is defined in the second paragraph of the Materials and Methods which corresponds to its first appearance. 

The bracket containing DMEM and FBS is unclear.

We have changed our phrasing, we hope the new wording is clear.

As for all other companies, the location and state of Dentaid should be named with the first appearance.

We have included the missing information the first time Dentaid is mentioned.

This sentence is extremely complicated and hard to understand and should therefore be simplified and shortened: "To generate this virus the RSV infectious clone system, obtained from BEI Resources (NIAID, NIH: Bacterial artificial chromosome plasmid pSynkRSV-I19F containing antigenomic cDNA from RSV A2-Line19F, NR-3646), was transfected into HEK-293 cells together with a codon-optimized T7 polymerase (Addgene, United States, 65974), and an RSV N, P, M2-1, and L expression plasmids at a ratio of 4:2:2:2:1."

We have changed the sentence. We hope it is clear now.

Discussion:

L.287: How do you define a "large study"? Please specify.

By “large” we meant the most comprehensive study. We have changed the text accordingly. 

Some of the new paragraphs in the discussion are lacking relevant references, especially 295-308. Please add.

We apologize if the statements in the discussion seemed unsupported by the literature. We have added references in those places in which we felt they were necessary.

---

## [Decision Letter · Decision Letter 2]

2 Jan 2024

Cetylpyridinium chloride and chlorhexidine show antiviral activity against Influenza A virus and Respiratory Syncytial virus in vitro

PONE-D-23-32284R2

Dear Dr. Luis Martinez-Gil,

We’re pleased to inform you that your manuscript has been judged scientifically suitable for publication and will be formally accepted for publication once it meets all outstanding technical requirements.

Kind regards,

Ajaya Bhattarai

Academic Editor

PLOS ONE

Additional Editor Comments (optional):

The revised manuscript looks good.

Reviewers' comments:

Reviewer's Responses to Questions

**Comments to the Author**

1. If the authors have adequately addressed your comments raised in a previous round of review and you feel that this manuscript is now acceptable for publication, you may indicate that here to bypass the “Comments to the Author” section, enter your conflict of interest statement in the “Confidential to Editor” section, and submit your "Accept" recommendation.

Reviewer #3: All comments have been addressed

2. Is the manuscript technically sound, and do the data support the conclusions?

Reviewer #3: Yes

3. Has the statistical analysis been performed appropriately and rigorously? 

Reviewer #3: Yes

4. Have the authors made all data underlying the findings in their manuscript fully available?

Reviewer #3: Yes

5. Is the manuscript presented in an intelligible fashion and written in standard English?

Reviewer #3: Yes

6. Review Comments to the Author

Reviewer #3: Dear authors,

thank you for taking my comments on board and congratulation for the interesting manuscript!

Best wishes

7. PLOS authors have the option to publish the peer review history of their article (what does this mean?). If published, this will include your full peer review and any attached files.

Reviewer #3: No

---

## [Editor Report · Acceptance letter]

8 Feb 2024

PONE-D-23-32284R2 

PLOS ONE

Dear Dr. Martinez-Gil, 

I'm pleased to inform you that your manuscript has been deemed suitable for publication in PLOS ONE. Congratulations! Your manuscript is now being handed over to our production team.

Kind regards, 

on behalf of

Dr. Ajaya Bhattarai 

Academic Editor

PLOS ONE